# High Quantum Efficiency and Broadband Photodetector Based on Graphene/Silicon Nanometer Truncated Cone Arrays

**DOI:** 10.3390/s21186146

**Published:** 2021-09-13

**Authors:** Jijie Zhao, Huan Liu, Lier Deng, Minyu Bai, Fei Xie, Shuai Wen, Weiguo Liu

**Affiliations:** School of Opto-Electronic Engineering, Xi’an Technological University, Xi’an 710021, China; zhaojijie@st.xatu.edu.cn (J.Z.); liuhuan@xatu.edu.cn (H.L.); denglier@xatu.edu.cn (L.D.); baiminyu@xatu.edu.cn (M.B.); feixie@xatu.edu.cn (F.X.); wenshuai@st.xatu.edu.cn (S.W.)

**Keywords:** graphene, Si NTCAs, quantum efficiency, anti-reflection, photodetectors

## Abstract

Light loss is one of the main factors affecting the quantum efficiency of photodetectors. Many researchers have attempted to use various methods to improve the quantum efficiency of silicon-based photodetectors. Herein, we designed highly anti-reflective silicon nanometer truncated cone arrays (Si NTCAs) as a light-trapping layer in combination with graphene to construct a high-performance graphene/Si NTCAs photodetector. This heterojunction structure overcomes the weak light absorption and severe surface recombination in traditional silicon-based photodetectors. At the same time, graphene can be used both as a broad-spectrum absorption layer and as a transparent electrode to improve the response speed of heterojunction devices. Due to these two mechanisms, this photodetector had a high quantum efficiency of 97% at a wavelength of 780 nm and a short rise/fall time of 60/105µs. This device design promotes the development of silicon-based photodetectors and provides new possibilities for integrated photoelectric systems.

## 1. Introduction

As one of the most important raw materials in the electronics industry, silicon is widely used in photoelectric detectors and semiconductor chips due to its low price and high performance [1]. Light loss is one of the main factors that affects the efficiency of photoelectric conversion in photodetectors. Most of the light that hits the surface of photodetectors cannot be absorbed by their optoelectronic material, but is instead scattered or converted into heat energy [2,3]. Therefore, researchers have investigated a wide range of methods for promoting the light absorption properties of Si-based photodetectors.

Using double-layer or multi-layer antireflection films is a traditional method for enhancing light absorption. Films with different refractive indexes are stacked on the surface of silicon wafers. This reduces reflectivity by lowering the refractive index mismatch between air and the silicon substrate [4]. However, antireflection films have wavelength and angle selectivity and only show good antireflection performance in specific wavelength and angle ranges. Light capture by structured surfaces is another effective method for promoting absorption [5,6]. Nanostructured materials with different dimensions, such as nanowires, nanotubes, nanopillars, quantum wells, quantum dots, and nanocrystals [7,8], have received increasing attention in recent years. Compared with bulk and planar Si structures, nanostructured Si materials with large surface-to-volume ratios, powerful light trapping performance, and excellent charge transport characteristics present great potential for photoelectric detection. Lu et al. constructed an ultrasensitive photodetector by integrating 2D In_2_S_3_ and a Si nanopillar array [9]. They explained that the enhancement of light absorption in 2D In_2_S_3_ was due to the Si nanopillars acting as Fabry–Pérot (FP)-enhanced Mie resonators.

As a zero-bandgap semiconductor, graphene has superior properties such as low resistivity, high electron mobility, and high mechanical strength. Its ability to absorb incident light from ultraviolet to infrared makes the realization of broadband photodetectors possible [10,11]. However, the responsivity of graphene-based photodetectors is strictly limited due to their extremely low absorption [12].

In order to combine the extraordinary properties of both structured Si and graphene, graphene and Si nanometer truncated cone arrays (NTCAs) were used to construct high quantum efficiency and broadband photodetectors in this paper. Compared with planar Si or Si nanopillars, NTCAs further enhance light absorption, confirmed by both simulation and experimental results in this study. On this basis, graphene was transferred as a transparent upper electrode and formed a heterojunction with silicon to prepare a high quantum efficiency photodetector. The photoelectric characteristics, including spectral response, responsivity, current-voltage characteristic, and response time, were all recorded and analyzed.

Herein, graphene and Si NTCAs were used to construct photodetectors with high quantum efficiency and broadband detection. Because of the light-trapping effect of the multipole FP-enhanced Mie resonant modes of periodic nanostructures, these Si NTCAs have a stronger absorption capacity for incident light [13]. This heterojunction-based photodetector can overcome the weak light absorption and severe surface recombination that are inevitable in traditional Si-based photodetectors. Meanwhile, graphene can be used both as a broad-spectrum absorption layer and as a transparent electrode to promote the response speed of the heterojunction device. As a result, the photodetector can work under zero bias with excellent photodetection performance. Moreover, the van der Waals (vdWs) heterojunction formed between the Si NTCAs and graphene both suppresses the dark current and promotes the separation of photogenerated carriers in the depletion zone, and this can improve the quantum efficiency of graphene/Si NTCAs devices. Passing the test, the quantum efficiency of the graphene/Si NTCAs device was as high as 97% at 790 nm. Due to the zero bandgap of graphene, the heterojunction photodetector showed a capability for broadband detection of ultraviolet (UV) to near-infrared (NIR) light (from 350 nm to 1550 nm), broadening the detection range of Si-based photodetectors. In addition, the response time (rise/fall time of 60/150 µs) of the constructed graphene/Si NTCAs device was in the order of microseconds, shorter than that of most other vdWs heterojunction photodetectors based on 2D layered materials. This graphene/Si NTCAs device reveals unique opportunities for future high quantum efficiency, broadband, and high-speed photodetectors.

## 2. Experiment

A schematic diagram of the obtained graphene/Si NTCAs heterojunction photodetector is shown in Figure 1. Vertically ordered Si NTCAs are prepared on an N-doped (resistivity 1–10 Ω·cm) Si wafer with a thickness of 500 µm by reactive ion etching (RIE). The prepared Si NTCAs samples were 1 × 1 cm^2^ in area. Then, 3–5 layers of 1 × 1 cm^2^ graphene (VIGON Technologies Co., Ltd., Hefei, China) was transferred onto the top of Si NTCAs by a polymethyl methacrylate (PMMA)-assisted process [14]. Silver electrodes were deposited on top of the graphene using silver paste (Shenzhen Sinve New Materials Co., Ltd., Shenzhen, China) by the screen-printing method. A graphene film and silicon substrate were used to collect photogenerated carbon carriers. The effective sensing area was 0.6 × 0.6 cm^2^, patterned by the silver electrode that has a width of 1 mm, and a length and height of 2 µm. To verify the excellent performance of Si NCTAs, we also prepared a graphene/Planar Si heterojunction photodetector, which has a similar structure to the graphene/Si NTCAs heterojunction photodetector device. In the entire device, we simply replaced the Si NTCAs with planar Si, and the preparation method was exactly the same as the above steps.

The morphology of the Si NTCAs and graphene/Si NTCAs was obtained by scanning electron microscopy (SEM; SU1510). The composition of graphene was characterized by Raman spectroscopy (HORIBA XploRA PLUS). Photodetector device performance was investigated with a semiconductor characterization system composed of a spectrometer (DSR-F4-XIAN, Zolix, Beijing, China), a precision source meter (Keysight B2901A, Santa Rosa, CA, USA), and a lock-in amplifier (Model SR830 DSP, Zurich, Switzerland).

## 3. Results and Discussion

SEM images revealed that the Si NTCAs are composed of vertically aligned truncated cones, as shown in Figure 2. The height of each silicon truncated cone is 800 nm and the diameters of the top and bottom sides of the truncated cones are 300 nm and 500 nm, respectively.

Figure 3 shows the Raman spectrum of the graphene layer on top of the Si NTCAs. The two peaks at 1582 cm^−1^ and 2700 cm^−1^ represent the characteristic G and 2D bands of graphene, respectively. These two peaks are sharp, and their intensity ratio (I_2D_/I_G_) is 1.42, meaning that the as-prepared graphene is a monolayer. The SEM image (inset in Figure 3) shows that the graphene film is uniform and continuous, demonstrating high quality [15,16].

In order to characterize the light trapping properties of the Si NTCAs, a finite differential time domain (FDTD) method-based simulation was conducted. The reflections of planar silicon and silicon nanopillars were also studied for comparison. The simulation model is shown in Figure 4. Silicon was chosen as the simulation material and a cuboid with a length of 1.5 μm, a width of 1 μm, and a height of 5 μm was established as the planar silicon model (Figure 4a). The height and diameter of the pillars in the nanopillar array model were 800 nm and 500 nm, respectively (Figure 4b). For the Si NTCAs model, the diameters of the top and the bottom sides of the truncated cone were 300 nm and 500 nm; the height of the cone was 800 nm (Figure 4c) [17].

The simulation results are presented in Figure 5a. Within the wavelength range of 400–1100 nm, the micro-structured silicon substrates (silicon nanopillars and Si NTCAs) showed significant light-trapping properties, leading to a reflection of less than 40% for most wavelengths. Compared with silicon nanopillars as discussed in the early work of other researchers [5,18], the Si NTCAs substrate in this work showed a better light trapping effect, with an antireflection improvement of up to 30% in the wavelength range of 400–1100 nm. Figure 5b shows the measured reflection values of the three different substrates, and these values are consistent with the simulation results. Only about 15% of incident light was reflected from the Si NTCAs substrate in the visible light region and 20% was reflected in the near-IR region. This can be attributed to the continuous gradient change in the Si NTCAs from the upper to the lower surface of the truncated cones. According to the effective medium theory [19,20], when light moves from air to silicon nano-circular platform arrays, this continuous gradient change reduces the sudden change in refractive index. Therefore, the Si NTCAs substrate has a better antireflection performance than the silicon nanopillar array and demonstrates enhanced multiple scattering.

The physical structure of a photodetector based on a graphene/Si NTCAs heterojunction is shown in Figure 6a. The physical structure of a photodetector based on a graphene/Planar Si heterojunction is shown in Figure 6b. In order to further investigate the electrical characteristics of the heterojunction, current–voltage (I–V) curves of graphene/Si NTCAs and graphene/Planar Si heterojunction devices were obtained in the dark and under 780 nm light irradiation (25 mW/cm^2^), and are shown in Figure 6c. Due to the formation of the heterojunction between graphene and the Si, both devices showed excellent rectification characteristics. The curves show that the dark current of the graphene/Si NTCAs device was much lower than that of the graphene/planar Si device. In contrast, the photocurrent of the graphene/Si NTCAs device was much higher than that of the graphene/planar Si device. Moreover, in the dark, the rectification ratio of the graphene/planar Si device is 8.1 and that of graphene/Si NTCAs is up to 20.4 within ±5 V. Compared with the current on/off ratio (I_on_/I_off_) of 5.2 obtained by the graphene/planar Si device at a bias voltage of 5 V, the graphene/Si NTCAs device performed with a higher on/off ratio, exceeding 100. It is believed that the contact area of graphene and Si and the high absorption of the Si NTCAs to incident light cause these differences.

External quantum efficiency (EQE) analysis is shown in Figure 6d, presenting a wide-band response in the range of 380–1100 nm for both heterostructure devices. However, the quantum efficiency of the graphene/Si NTCAs device is significantly higher than that of the graphene/planar Si device. The highest EQE of the graphene/Si NTCAs heterojunction device is 97% at 790 nm, twice as high as that of the graphene/planar Si device. This is because the periodic Si NTCAs have a stronger absorption than planar Si, as shown in Figure 5.

According to Equation (1) [21], the responsivity (R) of the graphene/Si NTCAs photodetector can be calculated through the measured EQE. The calculated R of the graphene/Si NTCAs device is 0.45 A/W at 900 nm, better than that of traditional inorganic diodes (0.3 A/W) [22]. These results are further proof that the Si NTCAs can be used as a Mie resonator and can effectively enhance light absorption.
(1)$$\mathrm{EQE}=\frac{J/e}{P/\left(hc/\lambda \right)}=\frac{hc}{e\lambda }\times \frac{J}{P}=\frac{hc}{e\lambda }\times R$$

In addition to responsiveness, an important measure excellent performance of the detector at detection rate, the detectivity (D*) can be expressed as [23]:(2)$$D^{*}=\frac{R_{\lambda }\cdot \sqrt{A}}{\sqrt{2{\mathrm{eI}}_{\mathrm{Dark}}}}$$
where Rλ denotes the responsivity, A is the effective area of the device, and I_Dark_ is the dark current. By testing at 780 nm, 0 V cesium, dark current I_Dark_ = 1.24 × 10^−6^ Hz^1/2^A, and photosensitive area A = 0.36 cm^2^, we were able to obtain R_λ_ = 0.45 A/W, so D* was 4.286 × 10^11^ cm Hz^1/2^ W^−1^.

When the incident radiation is weak, the generated signal current is equal to the noise current. When the signal is submerged in the noise, it cannot be resolved. The incident radiation power is the minimum power of the detector, thus known as the noise equivalent power (Noise Equivalence Power, NEP) [24].
(3)$$D^{*}=\frac{\sqrt{A\Delta f}}{\mathrm{NEP}}$$
where Δf is the electrical bandwidth, therefore the device NEP is 5.2 × 10^−12^ W.

By studying the response characteristics of the graphene/Si NTCAs heterojunction device under different voltages, stable, reversible, and repeatable optical responses with different Ion/Ioff ratios can be observed when the light (780 nm, 25 mW/cm^2^) is turned on and off, as shown in Figure 6e. The sensitivity to light increases with increasing the bias voltage from 1 V to 5 V, and I_on_/I_off_ values of 5, 20, 30, 55, and 104 are obtained at bias voltages of 1 V, 2 V, 3 V, 4 V, and 5 V, respectively. This indicates that this device can effectively generate and separate electron-hole pairs. Moreover, the representative response properties of the device to UV (365 nm) and NIR (1550 nm) light at a bias voltage of 3 V are shown in Figure 6f,g. The excellent switching stability, reproducibility, and significant photoresponse indicate that this graphene/Si NTCAs heterojunction photodetector has significant potential for application in broadband photodetection.

Another important property of photodetectors is response speed, which represents their ability to monitor quickly varying optical signal. Figure 7a–c show representative repeated switching of the photodetector between low and high frequency states. The frequency generated by the laser diode was periodically turned on and off at frequencies of 500 Hz, 5000 Hz, and 10 kHz. The graphene/Si NTCAs photodetector can operate with excellent stability and repeatability at a wide frequency range, up to 80 kHz. In addition, the relative balance of photocurrent [(I_MAX_ − I_MIN_)/I_MAX_] as a function of modulation frequency is shown in Figure 7d. The relative balance reaches 88.2% at a frequency of 10 kHz, indicating that the graphene/Si NTCAs photodetector has the ability to monitor ultra-fast optical signals.

The response time is usually defined by the rise time and the fall time, which are the time from 10% to 90% of the rising edge of the photodetector response curve and the time from 90% to 10% of the falling edge, respectively [25]. Response time is mainly affected by the capacitance and photogenerated carriers between the electrodes [26]. When light irradiates the surface of a device, the generated carriers are injected into the barrier area and the device quickly responds. With the increase in number and subsequent diffusion of carriers, the response of the device gradually slows down until it reaches stability. The rise time of the graphene/Si NTCAs device is 60 µs, as shown in Figure 7e. In contrast, when the incident light is turned off, the photoelectric voltage disappears and the response of the device in the barrier region rapidly drops. The fall time of the graphene/Si NTCAs device is 105 µs, as shown in Figure 7e. The response speed of this graphene/Si NTCAs device is two orders of magnitude faster than that of an In_2_S_3_/Si nanopillar device reported in reference [9]. This is because graphene has a much higher carrier mobility than In_2_S_3_ and can act as a transparent electrode in the fabricated graphene/Si NTCAs.

Light intensity is also a key factor in determining the photocurrent of a photodetector. The light intensity-dependent light response of the graphene/Si NTCAs device was further investigated, as shown in Figure 7f. When the power intensity of the incident light was increased from 8.3 µW/cm^2^ to 25 mW/cm^2^, the optical response was significantly enhanced. This section may be divided by subheadings. It should provide a concise and precise description of the experimental results and their interpretation, as well as the experimental conclusions that can be drawn.

Band diagrams and carrier migration schematics of the graphene/Si NTCAs device is shown in Figure 8, Due to the low work function of silver and the tunability of the Fermi level of graphene, the work function of graphene (P-type) is lower than that of silver in the air. Therefore, the reverse bias has no obstacle to the transfer of holes from graphene to silver. Although there is no ohmic contact between the two, this will not affect the hole transport of the device [27].

The graphene and silicon form a PN junction, and the built-in electric field, is directed from silicon to graphene, so that the device can work well under reverse bias voltage. Since the contact area between the graphene and the silicon nanopillar is small, that is, the area of the heterojunction region is small, the large contact resistance between the two has little effect on the response characteristics of the device, and it will bring a lower dark current. [28].

## 4. Conclusions

Si NTCAs were used as a light-trapping layer in combination with graphene to construct a high-performance graphene/Si NTCAs photodetector. This heterojunction-based photodetector can overcome the weak light absorption and severe surface recombination that are inevitable in traditional Si-based photodetectors. Meanwhile, graphene can be used both as a broad-spectrum absorption layer and as a transparent electrode to promote the response speed of the heterojunction device. As a result, the photodetector can work under zero bias with excellent photodetection performance. This photodetector device has a high quantum efficiency of 98% at 790 nm, broadband detection capabilities from the ultraviolet (UV) to near-infrared (NIR) regions (from 350 nm to 1550 nm), and a short rise/fall time of 60/150 µs [(I_MAX_ − I_MIN_)/I_MAX_], as a function of modulation frequency relative balance high reaches 88.2% at a frequency of 10 kHz, and 4.286 × 10^11^ cm Hz^1/2^ W^−1^ detection rate. The approach demonstrated herein advances the development of Si-based photodetectors and offers new possibilities toward integrated optoelectronic systems.

## Figures and Tables

**Figure 1 sensors-21-06146-f001:**
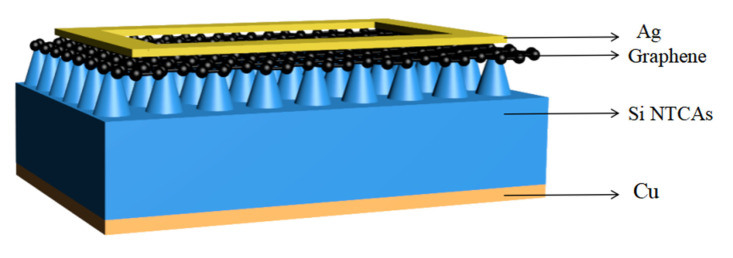
Schematic illustration of the graphene/Si NTCAs heterojunction photodetector.

**Figure 2 sensors-21-06146-f002:**
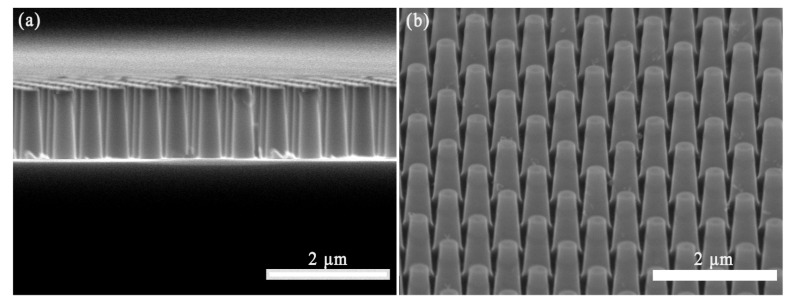
(**a**) Cross section and (**b**) top view SEM images of Si NTCAs.

**Figure 3 sensors-21-06146-f003:**
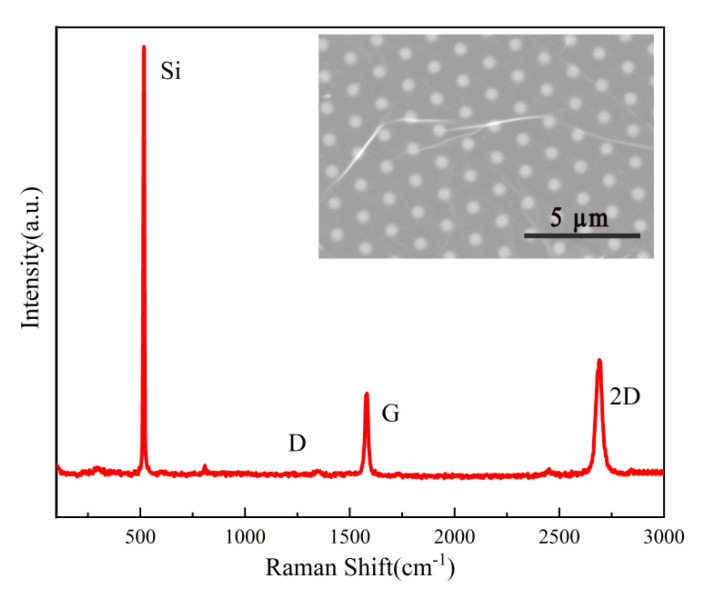
Raman spectrum of the graphene/Si NTCAs; the inset shows an SEM image of the graphene layer on the Si NTCAs.

**Figure 4 sensors-21-06146-f004:**
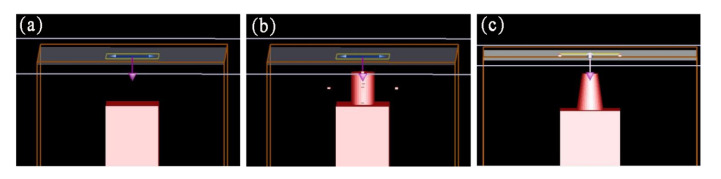
Simulation model of (**a**) planar silicon, (**b**) silicon nanopillars, and (**c**) Si NTCAs.

**Figure 5 sensors-21-06146-f005:**
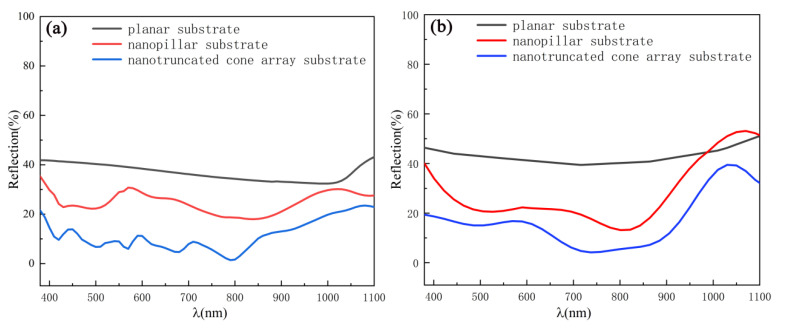
Reflections of planar, nanopillar, and nanometer truncated cone substrates obtained by (**a**) simulation and (**b**) experimental.

**Figure 6 sensors-21-06146-f006:**
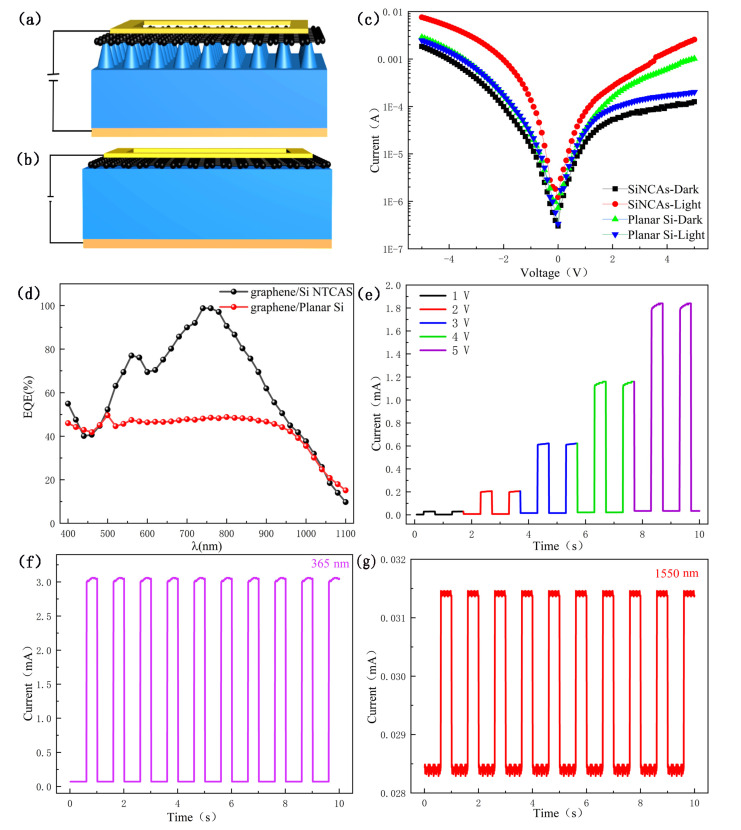
(**a**) Schematic diagram of graphene/Si NTCAs heterojunction device. (**b**) Schematic diagram of graphene/Plabar Si heterojunction device. (**c**) I-V curves of graphene/planar Si and graphene/Si NTCAs heterojunction devices in the dark and under light illumination (780 nm). (**d**) External quantum efficiency (EQE) of the graphene/Si NTCAs and graphene/Planar Si heterojunction devices. (**e**) Photoresponse of the graphene/Si NTCAs heterojunction under different bias voltages. Photoresponse of the graphene/Si NTCAs heterojunction under incident light of (**f**) 365 nm (36 mW/cm^2^) and (**g**) 1550 nm (45 mW/cm^2^) at a bias of 3 V.

**Figure 7 sensors-21-06146-f007:**
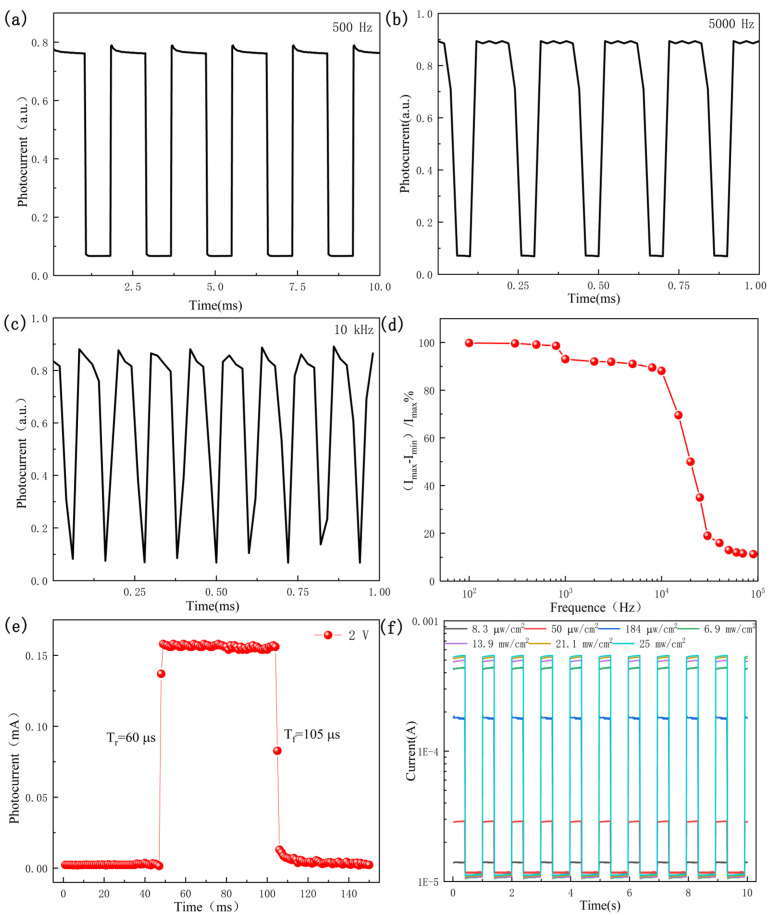
Photoresponse characteristics of the graphene/Si NTCAs heterojunction under pulsed light irradiation at frequencies of (**a**) 500 Hz, (**b**) 5000 Hz, and (**c**) 10 kHz at a voltage of 3 V. (**d**) Relative balance [(Imax-Imin)/Imax] vs. switching frequency. (**e**) Rising and falling edges for estimating rise time (τr) and fall time (τf). (**f**) Photoresponse of the graphene/Si NTCAs heterojunction under different light intensities.

**Figure 8 sensors-21-06146-f008:**
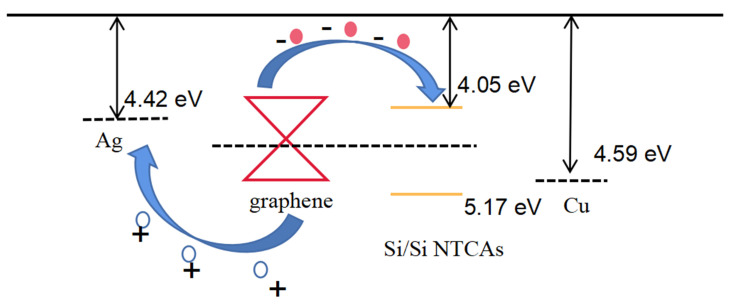
Band diagrams and carrier migration schematics.

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
