# Peer review of "High Quantum Efficiency and Broadband Photodetector Based on Graphene/Silicon Nanometer Truncated Cone Arrays"

_sensors, 2021, doi:10.3390/s21186146_

Round 1

Reviewer 1 Report

This manuscript presented a work about designing highly anti-reflective silicon nanometer truncated cone arrays (Si NTCAs) as a light-trapping layer in combination with graphene to construct a high-performance graphene/Si NTCAs photodetector. This photodetector has a high quantum efficiency of 97% at a wavelength of 780 nm and a short rise/fall time of 60/105 µs. The result is interesting. This manuscript is recommended for publication in Sensors after minor revision. The following issues should be addressed:

  1. The inset in Figure 3 is not clear, it is recommended to replace it with a clear SEM image.And Figure 6(d) is also not clear.
  2. There are some errors in the Figuue captions. For example, the repetitive use of (b) in the caption of Figure 5.
  3. There are spelling errors. For example, some superscript of the unit are  missingin the capton of Figure 6.
  4. Please indicate the film area of the Si NTCAs device and the thickness of each layer, because these parameters will have an impact on the performance of the device.
  5. In Fig. 6(b), the relative positions of the current valleys of the two devices have changed. Please explain the reason for this phenomenon.
  6. Please explain why the photoresponse of the graphene/Si NTCAs heterojunction is so stable under 1550 nm incident light(Fig.6(f)).
  7. Detection rate and noise equivalent power are the key factors that determine the performance of photodetectors. It is recommended to add relevant discussions. 

Reviewer 2 Report

The authors presented an interesting device structure having silicon NTCA and graphene cover and resulting data are interesting. However, this manuscript does not contain many details which must be addressed. Following are suggestions for improvement

1) Device fabrication process should be described in more detail. There are many questions remaining. Device size, electrode thickness and area, method of graphene growth, Raman spectrum of multiple points should be included.

2) The role of graphene is not clear.  Does it contribute to the reduction of reflection? There is no simulation including the graphene on silicon NTCA. Fig.4 and Fig.5 should include such data.

3) Effective areas of planar silicon and silicon NTCA are quite different. Device structure is quire different.  What is the device structure of planar device? If there is no pn junction, the planar device can only collect carrier from bulk which has a limited lifetime. The authors should include the discussion how the comparison between two structures can be fair?  

4) contact resistance between graphene /Ag and graphene/silicon NCTA top corner can be very high. Discussion on this aspect should be included.

Reviewer 3 Report

The manuscript is well written. The problem statement is presented with clarity and the theoretical/experimental procedure and the discussions flow well. The introduction serves to convince the readers about the need for this research and the experimental section is presented in a concise form.

Relevant background research has also been cited in the right places, but can be improved.

The manuscript can be published once the following minor improvements/issues listed below are taken care of:

1. FIGURE 5 is not visible in the pdf file. Please correct the pdf file and return to the reviewers (or at least the editor). The text accompanying the figure 5 did make sense.
2. Some more references can be added throughout the manuscript. For example, it will be helpful to add an example reference to the PMMA assisted process (Page 2, line 82).
3. Some numbers in the conclusion do not match the abstract. Please revise to make sure that the abstract, the results/discussion and the conclusion are consistent.
4. Consider expanding the introduction and conclusion to include all the important methods and results.
